# Patterns of Nighttime Crowd Flows in Tourism Cities Based on Taxi Data—Take Haikou Prefecture as an Example

**Bing Han** [1] , **Daoye Zhu** [1,2] , **Chengqi Cheng** [1,3] , **Jiawen Pan** [4,5] **and Weixin Zhai** [4,5,*]

1   Academy for Advanced Interdisciplinary Studies, Peking University, Beijing 100871, China; hanbing@stu.pku.edu.cn (B.H.); zhudaoye@pku.edu.cn (D.Z.); ccq@pku.edu.cn (C.C.)
2   Center for Geographic Analysis, Harvard University, Cambridge, MA 02138, USA
3   College of Engineering, Peking University, Beijing 100871, China
4   College of Information and Electrical Engineering, China Agricultural University, Beijing 100083, China; cau_panjiawen@cau.edu.cn
5   Key Laboratory of Agricultural Machinery Monitoring and Big Data Applications, Ministry of Agriculture and Rural Affairs, Beijing 100083, China
*   Correspondence: zhaiweixin@cau.edu.cn

**Abstract:** The study of patterns of crowd flows represents an emerging and expanding research field. The most straightforward and efficient approach to investigate the patterns of crowd flows is to concentrate on traffic flow. However, assessments of simple point-to-point movement frequently lack universal validity, and little research has been conducted on the regularity of nighttime movement. Due to the suspension of public transportation at night, taxi orders are critical in capturing the features of nighttime crowd flows in a tourism city. Using Haikou as an example, this paper proposes a mixed Geogrid Spatio-temporal model (MG-STM) for the tourism city in order to address the challenges. Firstly, by collecting the pick-up/drop-off/in-out flow of crowds, this research uses DCNMF dimensionality reduction to extract semi-supervised spatio-temporal variation features and the K-Means clustering method to determine the cluster types of nighttime crowd flows' changes in each geogrid. Secondly, by constructing a mixed-evaluation model based on LJ1-01 nighttime light data, crowd flows' clusters, and land use data in geogrid-based regions, the pattern of nighttime crowd flows in urban land use areas is successfully determined. The results suggest that MG-STM can estimate changes in the number of collective flows in various regions of Haikou effectively and appropriately. Moreover, population density of land use areas shows a high positive correlation with the lag of crowd flows. Each 5% increase in population density results in a 30-min delay in the peak of crowd flows. The MG-STM will be extremely beneficial in developing and implementing systems for criminal tracking and pandemic prevention.

**Keywords:** nighttime light data; crowd flows; taxi orders; urban land use area; DCNMF; K-Means



## 1. Introduction

The study of patterns of crowd flows represents an interdisciplinary and expanding research field [1]. Additionally, modeling and predicting human spatio-temporal movements also bears significant analytical value in urban environments, particularly in the post-epidemic era [2]. A crowd is defined as a group of people congregating in the same location together, which is generated from census data and represents the number of people who live in a region [3,4]. Crowd flows, on the other hand, refer to the movement patterns of groups over a period of time [5] and are primarily concerned with the quantity changes and flow variations in different land use areas [6]. Studies have shown that crowd flow data can assist urbanists and policymakers in mitigating traffic and planning for municipal public services, transportation resources, and other objectives [7,8]. The most direct and vital way to research crowd behavior dynamics is to study traffic flows [9,10].

However, our comprehension of human motion remains limited due to the lack of tools to track the time-resolved locations of individuals [11]. Significant variations in the sociodemographic characteristics of households, urban structure, industrial composition, and transportation systems have led to various changes in patterns of crowd flows in the last decade [12]. Vehicles equipped with a global positioning system (GPS) enable the efficient collection of vast amounts of movement data on individuals. However, raw data frequently lack activity information, which means that statistical processing is required [13]. Over the last decade, taxi floating car data have evolved into an important tool for investigating urban trip selection behaviors and activities [14]. With the development of the Internet, online car-hailing services, as an alternative to traditional taxi services, have been established, and they provide the advantages of efficient and high-speed data transmission and integration [15,16].

Relevant research based on taxi order data has made some progress. Kong [17] used taxi data to conduct time–location relationship research and effectively solved the empty taxi problem for drivers; meanwhile, the difficulty of finding a taxi for passengers was also reduced. Based on big taxi trajectory data, Li [18] completed related experiments and developed an experimental optimal taxi path model. Cartlidge [19] successfully obtained spatio-temporal predictions of shopping behaviors on the basis of taxi trajectory data. Xiong [20] proposed a topic model with a latent Dirichlet allocation algorithm to identify patterns of crowd flows and found that the travel distance in the morning rush hour was much shorter than that at other times. However, few studies have analyzed patterns of crowd flows combined with city land use data. Meanwhile, contrastive analyzing structures of crowd flows at citywide and downtown is a challenging task due to the dynamic changes generated by social activities [21], which has received scant attention in the research literature. In the analysis of intraurban crowd flows, simple point-to-point movement assessments often lack universal value. By considering the land use distribution in a city, the point-to-point taxi data can be partitioned, and the patterns of crowd flows can be assessed in different kinds of urban functional environments [22].

Additionally, few studies have focused on the behaviors and regularity of nighttime crowd flows. In urban environments, especially tourism cities, people engage in a variety of entertainment behaviors at night [23], which contrasts with their daytime mobility patterns. Therefore, adjusting the time at a certain stage can provide a more detailed representation of the city's population flow. Nighttime remote sensing sensors have unique advantages in human activity observation due to their exceptional sensitivity to visible light and near-infrared light where their sensitivity is several times that of ordinary sensors [24,25]. The nighttime image data from the Luojia 1-01 (LJ1-01) satellite have gradually matured, covering East Asia with higher accuracy than those from the Visible Infrared Imaging Radiometer Suite (VIIRS) [26]. By evaluating the intensity of nighttime light in different areas, it is feasible to estimate patterns of crowd flows and population distributions [27].

This paper uses Haikou which is a typical tourism prefecture-level city as an example to study patterns of nighttime crowd flows. In this work, according to the online car-hailing dataset, the mixed geogrid spatio-temporal model (MG-STM) in tourism cities is proposed to analyze the pattern of nighttime crowd flows in different land use areas. In this study, we partition Haikou prefecture into an $(I \times J)$ geogrids map based on the longitude and latitude. With collecting the pick-up flow, drop-off flow, and in-out flow of crowds, this paper uses the dual semi-supervised convex nonnegative matrix factorization (DCNMF) method for dimensionality reduction, which is a semi-supervised dimensionality reduction method for obtaining clear spatial-temporal features [28], then performs K-Means clustering method to acquire crowd night flows' changes cluster types in each geogrid. After presenting the spatio-temporal clustering, by constructing a mixed-evaluation model based on LJ1-01 nighttime light data, crowd flows' clusters, and land use data in geogrid-based regions, the patterns of all geogrid nighttime crowd flows are determined. Based on MG-STM, this paper analyzes the crowd flows' pattern changes aimed at different time periods and between downtown and citywide areas. Figure 1 depicts the research procedure. This

research will provide a valuable tool for forecasting crowd flows and managing urban functional areas.

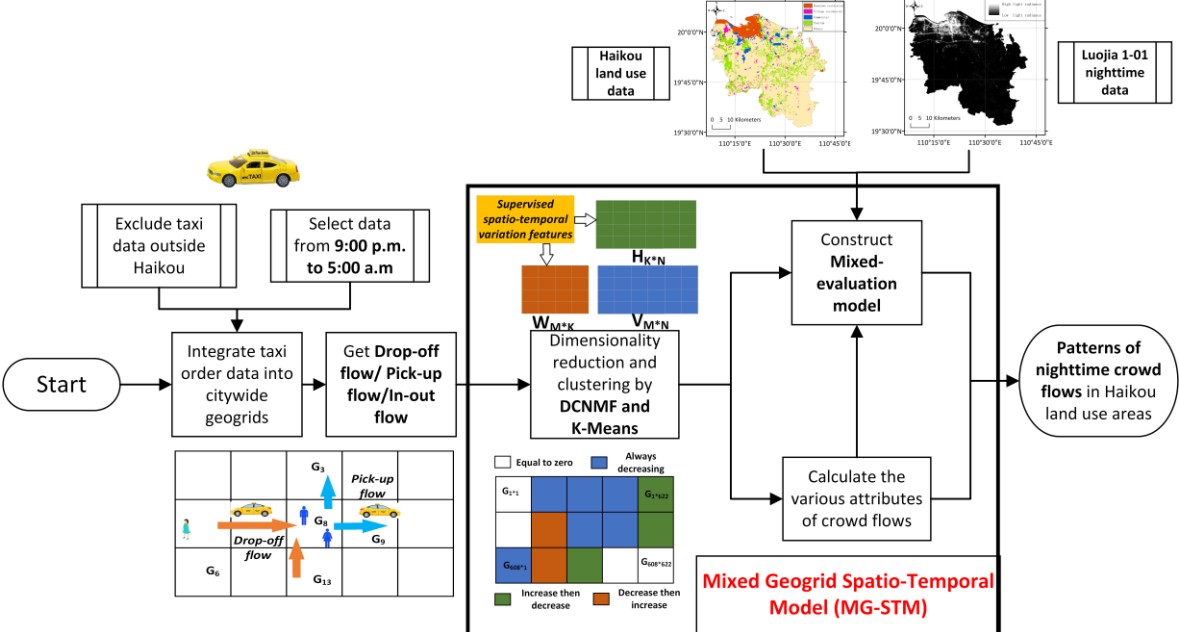

**Figure 1.** Research flowchart of patterns of nighttime crowd flows in Haikou based on taxi data.

## 2. Data

### 2.1. Land Use Data for Haikou

Haikou, whose permanent population is 2.2379 million people, is located at 19°31′N~20°04′N, 110°07′E~110°42′E on northern Hainan Island and is bounded by the Qiongzhou Strait. Haikou is known as a tourist destination due to its attractions such as Shishan Volcano Group National Geopark, Changqing Park, and Holiday Beach, which receives an average of 16 million visitors per year. The resolution of the obtained land use data for Haikou is 100 m.

Cities are complex systems fostering various kinds of land use areas (functional areas) [29]. According to the economic and development situation of Haikou city, the land use types are divided into five categories: Commercial, Tourism, Downtown Residential, Village Residential, and Other Areas. The subclass of each land use type is shown in Table 1. Therefore, Haikou has a diverse range of land use types, which results in various types of patterns of crowd flows in different land use areas. This work used geogrid-based processing to generate a map of Haikou's land use distribution, as illustrated in Figure 2a.

### 2.2. LJ1-01 Nighttime Light Data

Luminosity has informational value for countries with low-quality statistical systems, particularly for those countries with no recent population censuses [30,31]. Meanwhile, the rasterized bit map of nighttime light from VIIRS, such as Greater London, can also predicate grid-level population density distribution in London [32]. At the same time, the nighttime light intensity especially reflects the population density at night, which cannot be reflected by the ordinary population density map. To a certain extent, the nightlight remote sensing intensity can represent the nighttime population density at the geogrid-cell level, especially when only fixed-order ranking analysis will be performed [33]. Moreover, we conducted the experiment to compare the matching degree between the LJ1-01 nighttime light intensity and population density in Haikou, the results showed that the two-density data presents first-order linear matching relationship with $r^2 = 0.73$, which shows a highly linear positive correlation. The erroneous portion was due to old inaccurate population density data, and the nighttime light remote sensing imagery more accurately expresses the nighttime

information. Thus, by constructing a mixed-evaluation model that incorporates nighttime light data, taxi order changes in different areas, and land use data in geogrid-based regions, the patterns of nighttime crowd flows of a city can be determined.

The LJ1-01 satellite captures nighttime light imagery at 130-m resolution, which is higher than most existing nighttime light and population density images (1-km) to date [34], such as VIIRS Day/Night Band (DNB) data (1-km), which means smaller geogrids could be divided to achieve one-to-one coverage of the community by the geogrid. Furthermore, the LJ1-01 nighttime light imagery is preprocessed to remove the influence of irrelevant factors such as the amount of cloud cover. This study selects LJ1-01 nighttime light images of Haikou city on the evenings of 4 September, 20 September 2018, as shown in Figure 2b.

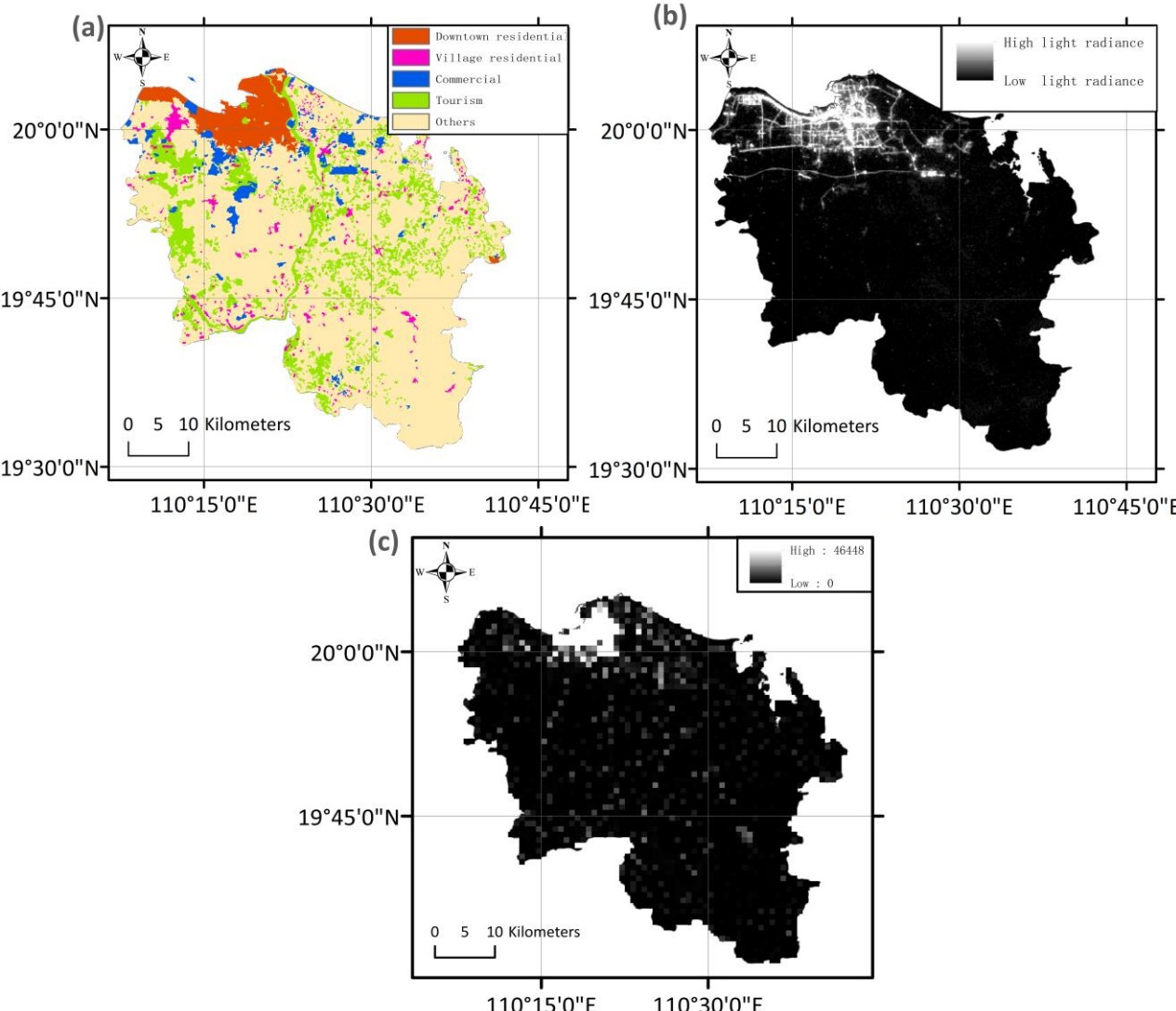

**Figure 2.** (**a**) Land use classifications in Haikou (2018). The map contains downtown residential, village residential, commercial, tourism and other land use types; each type is marked with a different color. (**b**) Nighttime remote sensing image of Haikou city captured by the LJ1-01 satellite (4 September 2018, as an example), the resolution of the map is 130 m. The image spans the latitudes and longitudes of 19°30′N to 20°15′N and 110°0′E to 110°45′E. In the map, the brighter the color, the higher the luminous intensity; the highest value is 973,851 W/(m²·sr·μm). (**c**) Population distribution map of Haikou (2010). The resolution of the map is 1 km. Population density and LJ1-01 nightlight remote sensing intensity shows a highly linear positive correlation, with R-square of 0.73.

**Table 1.** Subclass of each land use type.

| Land Use | Subclass |
|---|---|
| Commercial | business area and land for factories and mines, large industrial areas, oil fields, salt fields, quarries, as well as traffic roads, marine ports, airports and special land |
| Tourism | forests, meadows, sea beaches, tourist-type rivers, POIs, hotel clusters, and other areas conducive to the growth of tourism |
| Downtown Residential | the land in large, medium, and small cities and built-up areas above the county town |
| Village Residential | rural settlements, arable land and ranch land that are independent of towns |
| Other Areas | Sandy land, Gobi, saline land, marshland, bare land, bare rocky land, shrubland, reservoirs, and other unpopulated areas that are not suitable for developing secondary and tertiary industries |

*2.3. Taxi Data*

The sampling data are gathered every four hours to cover full peak and low traffic periods. The number of crowds entering and leaving the geogrid during this period is represented by Figure 3 in the form of a heat map. It is worth noting that during the morning rush hour, the density of people traveling from 5 a.m. to 9 a.m. is significantly lower than normal crowd flow in the heatmap [35]. The phenomenon occurs because, in addition to taxis, humans have a range of other modes of transportation during the day, including buses, shared bicycles, and so on. Therefore, taxi orders do not accurately reflect the intensity of daytime movement. However, due to the suspension of most buses and the unsafe riding of shared bicycles at night, individuals are increasingly inclined to choose taxis as a means of transportation. Consequently, in the middle of the night, from 9 p.m. to 5 am, the taxi information can correctly reflect the changes in the density of people.

Thus, this study selects the taxi order data of Didi Company for Haikou from 1 May to 31 October, each taxi order needs to satisfy the following two requirements: (1) Both the origin and destination of the order must be located in Haikou, and (2) the time of the order must be between 9 p.m. and 5 a.m. Taxi orders that do not meet the requirements will be removed. The reasons for selecting Didi data and this period to research patterns of crowd flows is:

- As a typical representative of a tourism city, Haikou has a large proportion of the mobile population (or called "tourists") and a low rate of private car ownership compared with conventional cities, especially in the peak tourism season selected for this paper. For the group of tourists studied in this research, car-hailing services become a valuable tool at night, facilitating their travel while extending the time of economic activity;
- Didi accounted for over 90% of all car-hailing services in China at the time of the statistics. Simultaneously, all cruising taxis in Haikou have been connected to and integrated with the Didi platform since 2017. Therefore, the taxi order data counted by Didi include dominated car-hailing taxis and traditional taxi data;
- According to information found from Haikou Government, most bus and shuttle services operate from 6 a.m. to 10 p.m. Additionally, due to known safety hazards, the number of people walking late at night is very rare and difficult to count, and the use of shared transportation at night is significantly reduced [36]. In summary, from 9 p.m. to 5 am, the evening peak hour has ended, and public transportation services have also almost been suspended; thus, taxi orders are the most accurate reflection of human night mobility characteristics.

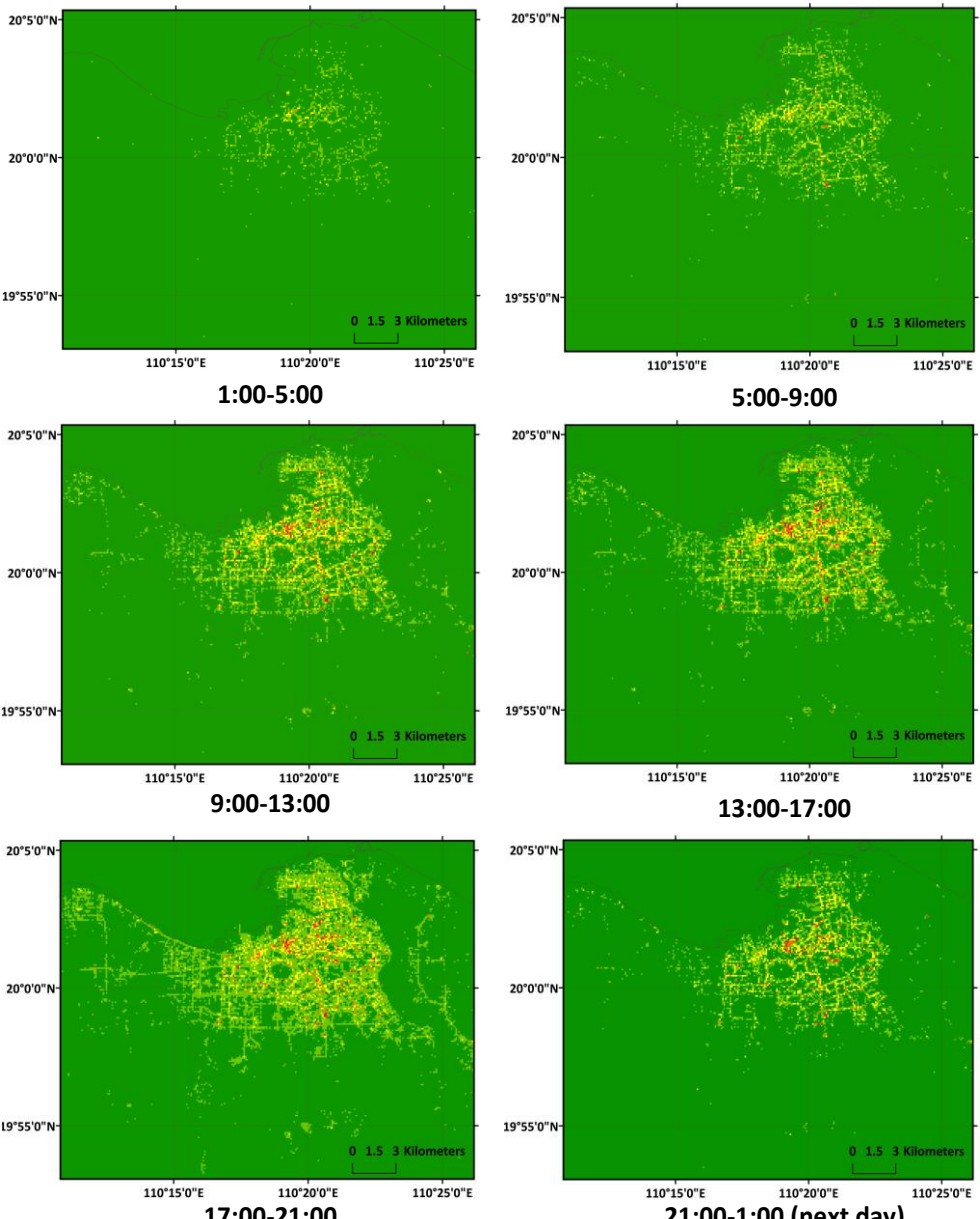

**Figure 3.** One–day heat map of collective mobility taking taxi. Set the sampling interval as 4 h, the brighter the color, the higher the mobility of people in that spatial–temporal geogrid.

Through the above analysis, the filtered Didi taxi order data effectively reflects the pattern of crowd flow at night. Although all data containing personal information has been anonymized, we can still infer the type of order, the number of passengers, the latitude and longitude of origin and destination, and the direction and distance from selected taxi data. Table 2 summarizes basic statistics about the datasets. In total, 2,058,131 observation orders from 9 p.m. to 5 a.m. are chosen to study nighttime and early-morning crowd flow in Haikou. By incorporating the geographic data of land use in Haikou with the location and time information from the taxi orders, it is possible to accurately determine the patterns of nighttime crowd flows for various land use types.

**Table 2.** Statistics of the taxi order datasets.

|  | Order 1 | Order 2 | Order 3 | Order 4 | ... |
|---|---|---|---|---|---|
| Order id | 175927190437 | 175928802315 | 175943512726 | 175958696894 | ... |
| Departure time | 19 May 2018 1:05:19 | 26 May 2018 0:02:43 | 23 July 2018 23:55:20 | 20 Sep 2018 21:49:00 | ... |
| Starting_lng, Starting_lat | 110.3665°, 20.0059° | 110.3249°, 20.0212° | 110.3446°, 19.9834° | 110.2913°, 20.0236° | ... |
| Arrive time | 19 May 2018 01:09:12 | 26 May 2018 00:04:47 | 23 July 2018 23:59:02 | 20 Sep 2018 21:53:17 | ... |
| Dest_lng, Dest_lat | 110.3645°, 20.0353° | 110.34629°, 20.0226° | 110.3598°, 20.0430° | 110.3740°, 20.0212° | ... |

## 3. Methods

### 3.1. Geogrid-Based Analysis

**Definition 1 (Geogrid).** *In this study, to analyze the changes of the taxi data in Haikou, the Haikou city space was divided into regular 608\*622 (I × J) geogrids with a 100-m spatial resolution based on the longitude and latitude. Each geogrid is assigned with one unique ID. All the geographic location information and order data for each geogrid are stored in the database, indexed by the unique ID.*

According to the geogrid latitude and longitude range, the order data are added to the corresponding geogrid database based on the positions of the order origin and destination [37]. During data processing, three types of database tables are created. Table records the number of taxis that travel in or out of the geogrid per hour. Figure 4a presents orders entering the geogrid in the "drop-off" table, which counts drop-off flow in time interval $t_{th}$. Orders leaving the geogrid are recorded in the "pick-up" table, as shown in Figure 4b, which counts Pick-up flow. In addition, Figure 4c presents the "in-out" table, which records the sum of the "pick-up" flow and "drop-off" flow.

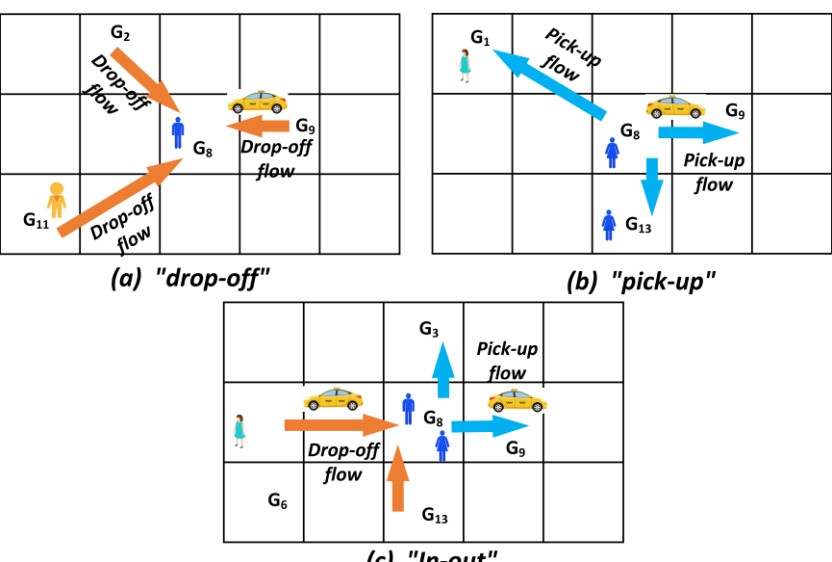

**Figure 4.** The three crowd flows in the city geogrid.

**Definition 2 (drop-off flow/pick-up flow/in-out flow).** *For a geogrid* $(i, j)$ *that lies at the ith row and the jth column, the drop-off flow, pick-up flow, and in-out flow of the crowds at the time interval t are defined respectively as*

$$x_{pick-up,i,j}^t = \sum_{t \in T} \sum_{r_k \in \mathcal{R}} \left\{ r_k^{origin} \in (i,j) \land r_k^{destination} \notin (i,j) \right\} \tag{1}$$

$$x_{drop-off,i,j}^t = \sum_{t \in T} \sum_{r_k \in \mathcal{R}} \left\{ r_k^{destination} \in (i,j) \land r_k^{origin} \notin (i,j) \right\} \tag{2}$$

$$x_{in-out,i,j}^t = x_{pick-up,i,j}^t + x_{drop-off,i,j}^t \tag{3}$$

*where R is the all-taxi orders in the time interval* $t_{th}$*,* $r_k$ *is* $k_{th}$ *certain sample in order.* $r_k^{destination} \in (i,j)/r_k^{origin} \in (i,j)$ *means the destination point/origin point of this taxi order lies within geogrid* $(i,j)$*, and vice versa. As for time interval* $t_{th}$*, because from 9 p.m. to 5 a.m. are chosen to research, the research time set T is denoted as Equation (4):*

$$T = \{t = 1_{st}, \ldots, 8_{th} | 21:00 - 22:00, 22:00 - 23:00, \ldots, 4:00 - 5:00\} \tag{4}$$

Figure 4 shows the schematic of the crowd flows in the three tables. At the $t_{th}$ time interval, we also define the three geogrid tables as a crowd flow tensor $X_t \in \mathbb{R}^{3 \times I \times J}$, where $(X_t)_{0,i,j} = x_{pick-up,i,j}^t$, $(X_t)_{1,i,j} = x_{drop-off,i,j}^t$, $(X_t)_{2,i,j} = x_{in-out,i,j}^t$, which denotes drop-off flow, pick-up flow and in-out flow in all 608*622($I \times J$) geogrids. Over a spatial geogrid represented, crowd flow Tensor $X_t$ could represent 3 types crowd flows change in each geogrid.

*3.2. Spatial-Temporal Dimensionality Reduction and Clustering of Crowd Flows*

3.2.1. DCNMF Dimensionality Reduction

After recording the crowd flow tensor in the geogrids, NMF dimensionality reduction is performed to extract spatio-temporal features based on the matrix [38]. NMF, as a widely used approach for dimension reduction, has the advantages of fast decomposition and high resolution. It is especially suitable for large-scale data analyses [39]. This paper adopts Dual Semi-Supervised Convex NMF (DCNMF) to completely exploit the limited label information to obtain spatial-temporal features, which outperforms the related state-of-the-art NMF methods [28]. We begin by constructing the data matrix based $V_{M \times N}$ on crowd flow tensor $X_t \in \mathbb{R}^{3 \times I \times J}$, which is expressed in Equation (5):

$$V_{M \times N}^{class} = V_{T \times (i \times j)}^{class} = [(X_{t=1_{st}})_{class}(:), \ldots, (X_{t=8_{th}})_{class}(:)]_{8 \times (608 \times 622)}^T, \tag{5}$$

where $class, = 0, 1, 2$, denotes "drop-off" table, "pick-up" table and "in-out" table respectively. $(:)$ *represents* the matrix would arrange the elements in a column order $(1 \times (I \times J))$.

In the data matrix $V_{M \times N}^{class}$, the first row represents the unique ID number of each geogrid, which is sorted from southwest to northeast; the second-to-last row represents the total number of people entering or leaving the geogrid in each hour. Through this geogrid-based method, the geogrid database in the three flows stores the collective flow changes per hour.

Then, DCNMF decomposes data matrix $V_{M \times N}$ into base matrix $W_{M \times K}$ and coefficient $H_{K \times N}$. We have:

$$V_{M \times N} = W_{M \times K} \times H_{K \times N} = V_{M \times N} U_{N \times K} H_{K \times N} = VU(AZ)^T \tag{6}$$

where $W = VU$, $H = (AZ)^T$, $A \in \mathbb{R}_{\geq 0}^{N \times (N+C-L)}$ is a label constraint matrix.

**Definition 3 (label constraint matrix).** *Previous studies have shown that the performance of algorithms can be improved by using limited supervised information [40], assume that the first L*

*data samples are labeled, and the rest of data samples are unlabeled. Each data sample has one class and the labeled samples have C classes in count, so the matrix A is:*

$$A = \begin{bmatrix} C_{L \times C} & 0 \\ 0 & I_{N-L} \end{bmatrix} \tag{7}$$

*where $C_{L \times C}$ is an $L \times C$ indicator matrix with $c_{ij} = 1$ if $x_i$ belongs to the jth class, otherwise $c_{ij} = 0$. In this paper, we denote L = 100, C = 4 in "pick-up", "drop-off" and "in-out" table, specific C class types and its meaning will be listed in Section 3.2.2.*

The commonly used objective function is still based on Euclidean distance, so the objective function for the constrained convex DCNMF is derived:

$$\min = \min_{W,H} \| V - WH \|_F^2 = \min_{U,Z,A} \sum_{i=1}^{M} \sum_{j=1}^{N} \left( V_{ij} - \left( VUZ^T A^T \right)_{ij} \right)^2 \tag{8}$$

We directly derive the multiplicative update rules of DCNMF, which is expressed in Equations (9) and (10):

$$u_{ik} \leftarrow u_{ik} \sqrt{\frac{(M^+ AZ + M^- UZ^T A^T AZ)_{ik}}{(M^- AZ + M^+ UZ^T A^T AZ)_{ik}}} \tag{9}$$

$$z_{ik} \leftarrow z_{jk} \sqrt{\frac{\left( A^T M^+ U + A^T AZU^T M^- U + \beta A^T \widetilde{S} AZ \right)_{jk}}{\left( A^T M^- U + A^T AZU^T M^+ U + \beta A^T \widetilde{D} AZ \right)_{jk}}} \tag{10}$$

where $M = V^T V$, $M^+ = \frac{|M|+M}{W}$, $M^- = \frac{|M|-M}{W}$, $\widetilde{S}$ is the new weight matrix, which is constructed by using Constraint Propagation Algorithm (CPA) that propagates the obtained pairwise constraints to the entire data. $D$ is a diagonal matrix with $D_{ii} = \sum_{j=1}^{N} \widetilde{s}_{ij}$.

According to Equations (9) and (10), the matrices $U$ and $Z$ are iterated sequentially. When the error is less than the threshold, we end the iteration, and the required matrices $W$ and $H$ are outputted. By introducing semi-supervised label constraint matrix $A$ DCNMF obtained the dimension reduction matrix $H$ with spatio-temporal variation features based on the pretreated labeled spatio-temporal class information. The complete DCNMF is illustrated in Algorithm 1.

---

**Algorithm 1.** Dual Semi-Supervised Convex NMF (DCNMF) algorithm.

Input: Data matrix $H$, label constraint matrix $A$, parameter $K = 4$.
Output: a base matrix $W_{M \times K}$ and a coefficient matrix $H_{K \times N}$.
1: Initialize matrices $W$ and $Z$;
2: Construct the weight matrix v by using CPA;
3: Update $U$ by using (9);
4: Update $Z$ by using (10);
5: If not Convergence, return step (3);
6: $W = VU$, $H = (AZ)^T$.

---

### 3.2.2. Clustering of Crowd Flows' Changes Based on Geogrids

After completing the dimensionality reduction, the base matrix $W_{M \times K}$ has obtained the temporal relationship of each geogrid, and sub $M$ represents the number of time interval steps. The coefficient matrix $H_{K \times N}$ represents the spatial dimension, and sub$N$ is the number of geogrids. Therefore, the purpose of dimensionality reduction has been achieved by using $H_i$ to represent geogrid $i$.

In this study, we denote $K = C = 4$, which means that the clustering of the geogrid will yield four categories. In clustering, geogrid cells are classified with similar spatio-temporal features into a cluster. Through DCNMF, each column of $H_{K \times N}$ is the normalized crowd flows' change eigenvalues generated after supervised learning. Therefore, according to the efficient matrix $H_{K \times N}$, this paper selects the K-means method that has good clustering effect on large data sets to cluster the changes of nighttime crowd flow in each geogrid [41], the completed K-Means algorithm is summarized in Algorithm 2.

---

**Algorithm 2.** Dual Semi-Supervised Convex NMF (DCNMF) algorithm.

---

Input: coefficient matrix $H_{K \times N}$, label constraint matrix $A$, cluster types $K_{means} = 4$.

Output: geogrid cluster type matrix $\Gamma \in \mathbb{R}^{3 \times I \times J}$ ("pick-up", "drop-off" and "in-out" table).

1: Select $K_{means}$ samples of different types in $A$ as the initial clustering center, $a = a_1, a_2, \ldots, a_{K_{means}}$;

2: For each sample $i_{th}$ column $H_{:,i}$ in $H_{K \times N}$, calculate the distance from sample to the $K_{means}$ cluster centers;

3: divide each sample $H_{:,i}$ into the class corresponding to the cluster center with the smallest distance;

4: For each cluster type, recalculate its cluster center $a_j = \frac{1}{|c_i|} \sum_{H_{:,i} \in c_i} H$;

5: If not Convergence, return step (2);

6: If not complete three tables, return step (1);

7: Output geogrid nighttime crowd flows' changes cluster type matrix $\Gamma$ in three flow tables.

---

Based on the crowd flows' changes in the geogrid cells per hour, four kinds of geogrid types are obtained by DCNMF and K-Means clustering: "remain unchanged (equal to zero)", "always decreasing", "increase then decrease", and "decrease then increase", which is shown in Figure 5. According to the four clustering results, four corresponding numerical trends in each geogrid are summarized. The four geogrid types represent the changes in the number of people entering or leaving the geogrid from 9 p.m. to 5 am, which correspond to the four cluster types.

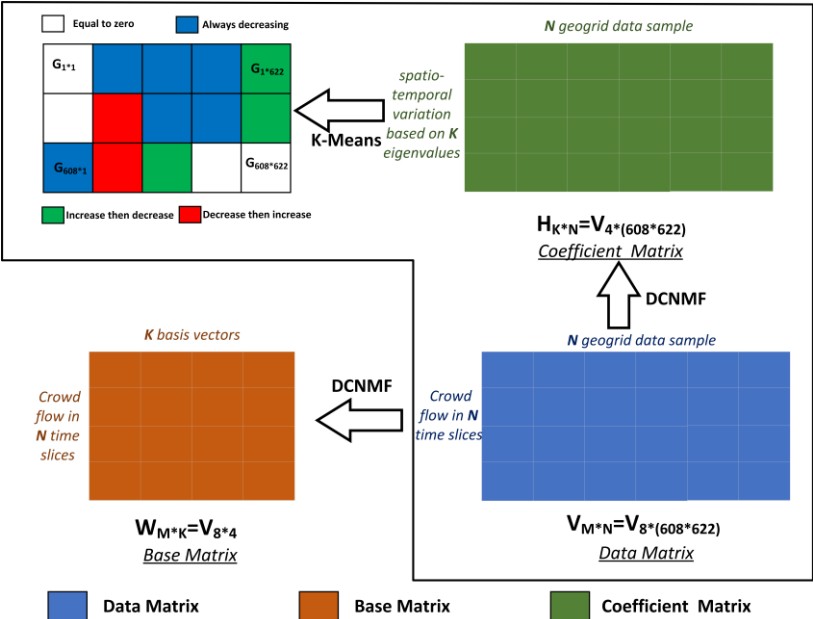

**Figure 5.** The spatial-temporal dimensionality reduction and clustering of crowd flow tensor $X_t$. The data matrix $V_{M \times N}$ divided into a base matrix $W_{M \times K}$ and a coefficient matrix $H_{K \times N}$ by means of DCNMF. By K-Means, geogrid nighttime crowd flows' changes cluster type matrix $\Gamma$ in three tables based on $H_{K \times N}$.

3.2.3. Methodological Evaluation

In dimensionality reduction and clustering of large spatio-temporal data, the geogrid type can reflect the collective flow in a region, but a single geogrid may not fully reflect collective trends. For example, "remain unchanged (equal to zero)" means that the geogrid crowd does not move at night, but when sporadic travel occurs, the geogrid may still be divided into a "remain unchanged" area. "Increase then decrease" means that the number of persons in the geogrid increases first and then decreases at night, but there may still be a sudden change of collective flow in a certain period of time. Thus, if the geogrid trend is most similar to this trend, it is classified as "increase then decrease". Each geogrid has its taxi change type in the three tables ("in-out" table, "pick-up" table, and "drop-off" table). This paper analyzes the clusters of changes in human nighttime mobility according to the geogrid location and obtains the change statistics map shown in Figure 6.

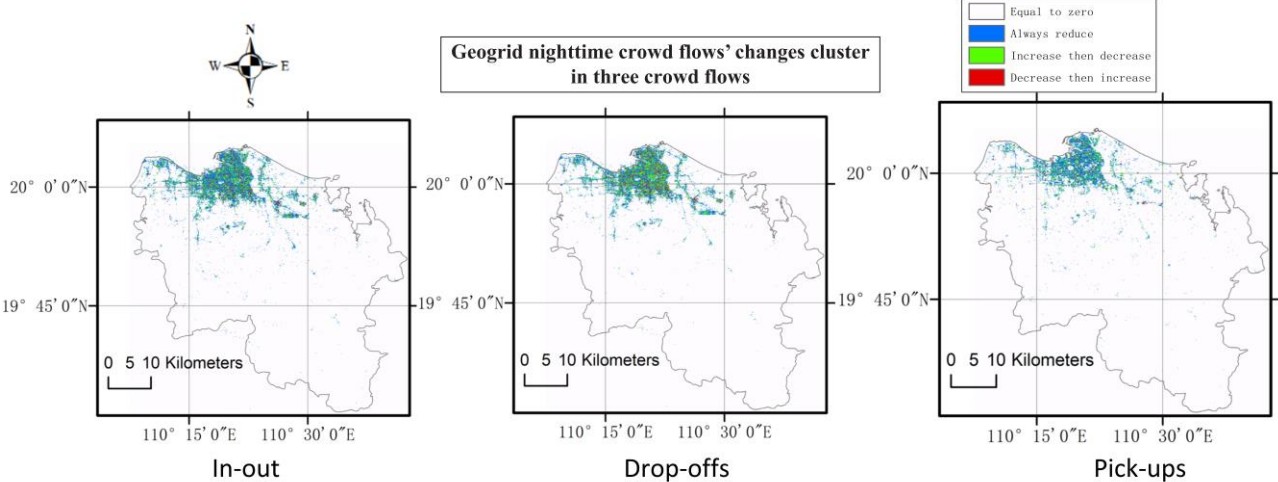

**Figure 6.** Crowd flows' changes in each geogrid at night in Haikou. The clustering results of crowd flows in the city are distinctive according to three different crowd flows.

Based on MG-STM, this paper will then analyze the patterns of crowd flows based on cluster types combined with Haikou land use and LJ1-01 nighttime data. Following the acquisition of LJ1-01 nighttime imagery, three different forms of mixed-evaluation model data are coupled via geogrid cells. Through the links among geographical regions, this paper analyzes the attributes of clustered geogrids, obtains the proportions of humans that take taxis for each considered land use type, and produces an LJ1-01 nighttime light intensity map.

## 4. Results

### 4.1. Statistical Analysis of Taxi Flows with Land Use Geogrid

With MG-STM, the proportions of crowd who take taxis in various land use geogrids are calculated based on the three nighttime crowd flows' changes cluster types in each geogrid, as shown in Figure 7. According to Figure 7, except for the downtown residential area, the percentage of the "always decreasing" geogrid is always the highest in the other land use types, with the proportions being above 75%. Comparing the two types of residential areas, the downtown residential area has a higher population density, while the village residential area is more dispersed and has a lower population density. Therefore, although the two areas are classified as residential, the types of patterns of crowd flows are different. In downtown residential areas, the proportion of "always reduce" geogrid cells exceeds 40%, and the proportions of "increase then decrease" geogrid cells and "decrease then increase" geogrid cells are also higher than those for the other land use types. In village residential areas, 85% of the area has approximately no activity at night.

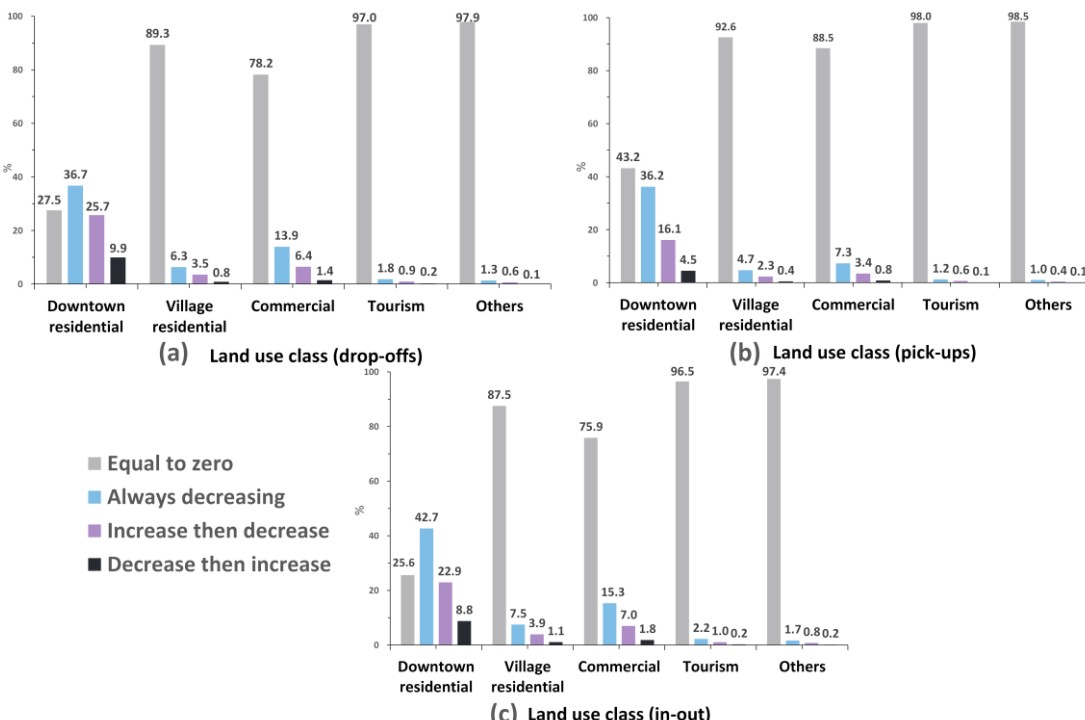

**Figure 7.** Proportions of crowd flows' change types in various land use geogrids based on the three taxis flows, (**a**–**c**) display the flow clustering results of three geogrid base flows (drop-off/pick-up/in-out flow of crowds).

In the commercial zone, the proportion of the "always reduce" and the "increase then decrease" geogrid cells is unusually relatively high, which is closely related to workers' preference to work in the commercial zone. For the "other" areas, which are sparsely populated, the most common crowd flows' pattern is "remain unchanged" ("equal to zero"), which indicates no population movement.

The differences of drop-off flow and pick-up flow are also compared. The results show that the "drop-off" area is more widely distributed than the "pick-up" area, which is approximately 35% larger than the area where people are "picking-up". This observation is also consistent with the pattern in people leaving work or recreational areas and returning to more dispersed residential areas. This phenomenon has led to the fact that in each land use type and time interval, the population number of pick-up flows keeps more than that of drop-off flows.

### 4.2. Changes in Nighttime Patterns of Crowd Flows Aimed at Different Time Periods

The flows of people in different time periods are also analyzed in the evening. Each night is divided into two periods: before midnight (9 p.m.–1 a.m.) and after midnight (1 a.m.–5 a.m.). The clustering results of crowd flows downtown movement are displayed in the form of spatio-temporal geogrids, as shown in Figure 8.

The results indicate that crowd flows were more frequent "before midnight" and that the size of the area with crowd flows changes was approximately 1.3 times larger than that in the "after midnight" period. Combined with geographic information and land use data, the more entertaining and bustling a certain area of the city is, the higher the probability of the crowd flows clustering becoming "increase then decrease" and "decrease then increase" in that geogrid area. One point worth noting is that the number of "decrease then increase" geogrids in the "before midnight" is greater than in the "after midnight" period, which was determined by the tourism characteristics of Haikou, such as people returning to hotels from entertainment venues and evening tourism areas, especially in the residential and tourism areas [42].

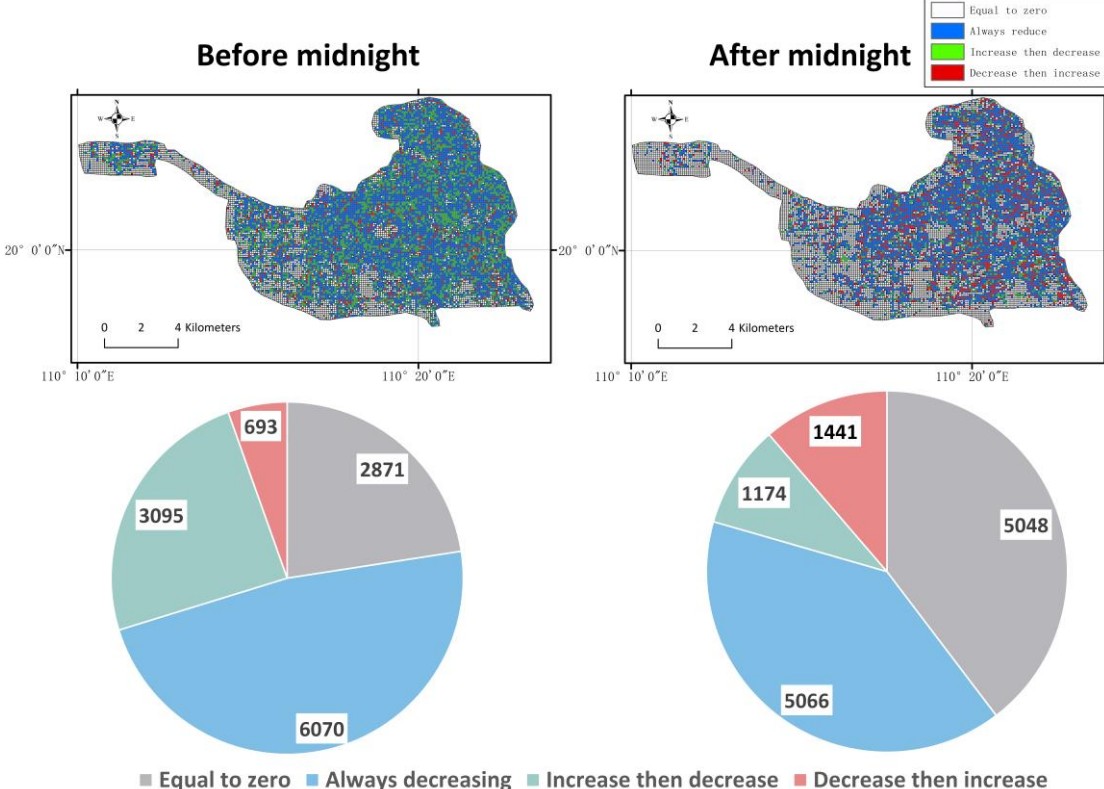

**Figure 8.** The distribution diagram of crowd flows' changes before and after midnight in downtown Haikou. The left side shows the geographical distribution and clustering geogrid quantity statistics for the crowd flows' changes in "before midnight", and the right side shows the crowd flows' changes in "after midnight".

### 4.3. A Comparison of Citywide and Downtown Crowd Flows

In order to summarize the densely populated areas in the urban core area with a targeted manner, we take the central urban area of Haikou as the research object, and collect corresponding data from the "China Haikou Yearbook (2018)" accordingly. The downtown part of Haikou is composed of 4 districts: Xiuying, Longhua, Qiongshan, and Meilan. We use the central urban area as the boundary to zoom in on the clustering results of crowd flow changes and display the clustering geogrids in the downtown area, as shown in Figure 9.

Compared with those in the citywide area, the collective flows are more frequent and larger in the downtown area, and people are active in most downtown places. In this paper, statistics are calculated on the clustering results for both maps. In the urban downtown district, the area where human movement is observed in 78.66% of the corresponding geogrids, the geogrids of the "increase then decrease" and "decrease then increase" clusters significantly increase.

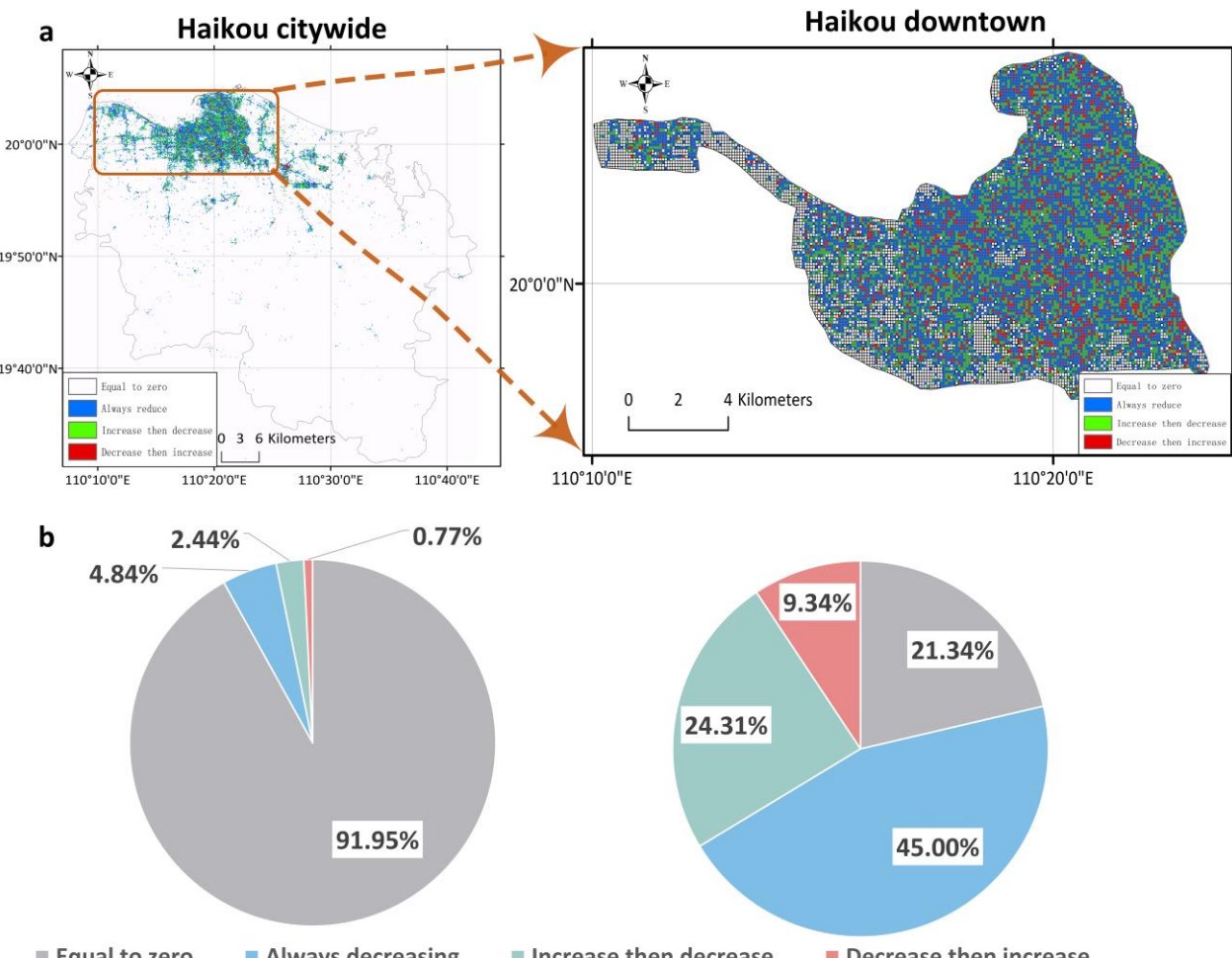

**Figure 9.** Distribution of crowd flow clustering in Haikou. (**a**) Maps of crowd flow clustering at night in citywide and downtown Haikou (based on in-out flow). Each unit geogrid corresponds to the change in people flow at night. Compared with the entire city-level map, there are more intensive taxi orders in downtown areas. (**b**) Corresponding to (**a**), clustering results of the human flow patterns between the citywide and downtown areas in Haikou.

### 4.4. Influence of the Nighttime Light Intensity on the Pattern of Crowd Flows

With the combination of land use data and LJ1-01 nighttime data, the ratio of nighttime light intensity for each land use type is calculated. Table 3 shows the percentages of the different nighttime light intensities in each land use area. According to the statistics of the geogrid graph, the nighttime light density in downtown residential areas is highest, and high-density areas are distributed in these areas, followed by commercial areas, village areas, and tourism areas, we have a fixed-order rather than quantitative description of population density in land use, which greatly reduces the error from nighttime light intensity to nighttime population density. The nighttime light intensity of the city is positively correlated with the population density [34], so we can also obtain the population distribution of each land use area. The urban area is divided into three types of population density areas according to the Jenks natural breaks method [43].

**Table 3.** Population densities of the different land use areas in Haikou city according to the percentage of nighttime light intensity.

| Land Use Area | Low Nighttime Light Intensity | Medium Nighttime Light Intensity | High Nighttime Light Intensity | Nighttime Population Density |
|---|---|---|---|---|
| Downtown residential | 16.53% | 27.94% | 55.52% | High population density (70–100%) |
| Commercial | 68.21% | 21.04% | 10.75% | High population density (70–100%) |
| Village residential | 88.80% | 9.91% | 1.30% | Medium population density (60–70%) |
| Tourism | 96.54% | 2.97% | 0.48% | Low population density (0–60%) |
| Other | 97.22% | 2.43% | 0.35% | Low population density (0–60%) |

The population density is highly correlated with the movement of taxis into and out of a certain location. Based on the foregoing conclusions, the mixed-evaluation model of nighttime light data, the crowd flows' clusters, and land use data in geogrid-based regions are used to analyze patterns of night crowd flows, as shown in Figure 10. In Haikou, the crowd who take taxis are concentrated in high-density areas, that is, urban residential areas and commercial areas. In the "equal to zero, "always decreasing", "increase then decrease" and "decrease then increase" areas, the proportions of areas with high population densities gradually increases. In other words, if the crowd flows cluster at night in an area is "decrease then increase", the probability that the land use area is downtown entertainment residential and the population density is high will be greatest.

Thus, when the crowd flow change cluster is "decrease then increase", the area is commonly developed, especially in downtown areas; this phenomenon also corresponds to the fact that in developed regions, due to overtime and entertainment, the collective flow generally decreases to less than in common regions [44]. According to the statistical results, the greater the population density is (which is positively correlated with the nighttime light intensity), the greater the probability that the nighttime population change cluster will be "decrease then increase", followed by "increase then decrease" and "always decreasing". The study finds that as the population density of the land use area increases, the regional crowd flows will lag accordingly. Within Haikou, for every 5% increase in population density, the peak of crowd flows will be delayed by 30 min, as shown in Figure 11.

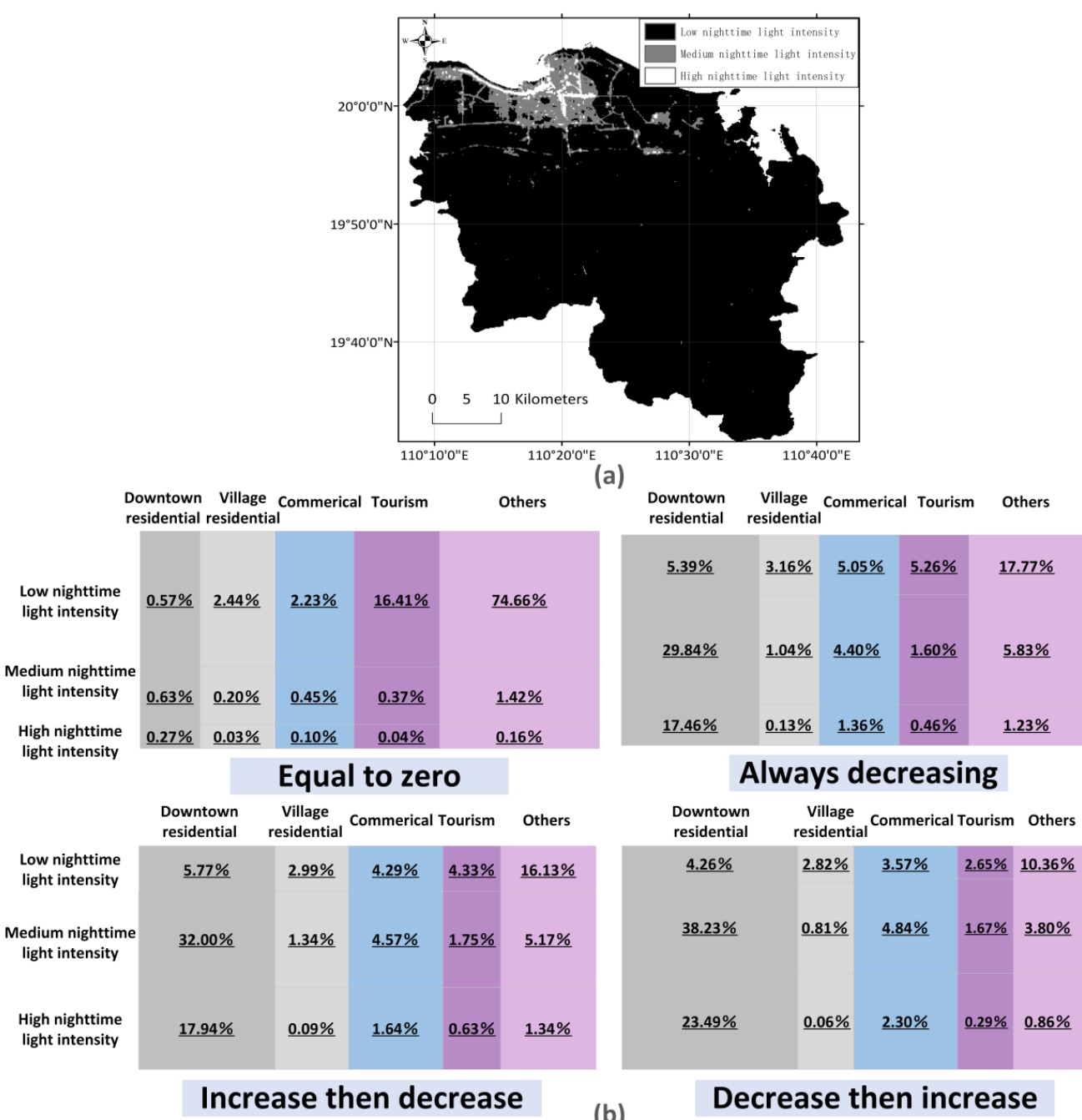

**Figure 10.** (**a**) Classification results of the nighttime light data in Haikou city after applying the Jenks natural breaks method. We select the nighttime light graded image on 20 September 2018, for display. Based on (**a**,**b**) which shows the proportions of geogrid with different nighttime light intensities (population densities) and land use types in the four crowd flow clusters. (**b**) includes the average proportions of two-day nighttime intensities.

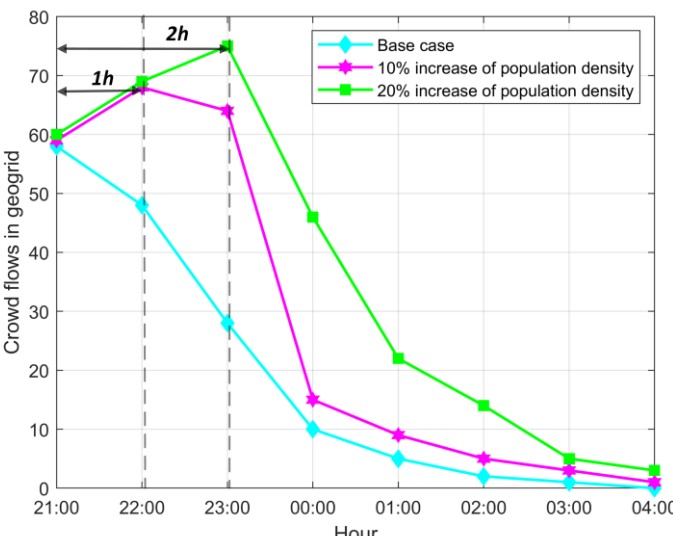

**Figure 11.** Correlation between night crowd flows and population density in Haikou. As the population density of the land use area increases, the regional crowd flows will lag accordingly. For every 5% increase in population density, the peak of crowd flows will be delayed by 30 min.

## 5. Discussion

Taxis are an important tool for citizens to travel, and their travel data can effectively reflect the flow pattern of people in the urban environment. By constructing MG-STM based on the taxi order information, patterns of nighttime crowd flows in urban land use areas can be analyzed.

Downtown residential areas have the highest population density and the highest nighttime collective traffic. The change clusters in crowd flows mainly fall into three types: "always decreasing", "increase then decrease", and "decrease then increase". Additionally, as areas become more prosperous, their population density increases and their crowd flow clusters shift toward "decrease then increase" and "increase then decrease," particularly "decrease then increase." Commercial areas are similar to those of downtown residential districts, and their population densities are relatively high. However, due to the concentration of commercial districts and fixed locations for taxis to pick up and drop off riders, there will be some areas with no movement of people. At night, the crowd flows clusters are mainly "always decreasing" and "increase then decrease" in commercial areas.

Although village residential areas are also important population agglomerations, their population densities are lower than commercial areas and downtown residential areas due to their geographical location and the abundance of upland rice fields. Expectations for nighttime suburban travel are low, and most of its areas are largely unpopulated. Additionally, in areas with population movement, the change in the collective flow is mainly "always decreasing". Tourism and other fields are large and sparsely populated, where most people do not have much activity at night. As Haikou is rich in tourism resources, there will be a certain increase in the population flow in the tourism area compared to that in other areas.

In summary, information on travel flows between areas would be important. Based on the mixed geogrid spatio-temporal model (MG-STM), the pattern of crowd flows in each area can be estimated. MG-STM first uses DCNMF and K-Means to establish nighttime crowd flow changes cluster of three types at the geogrid level through the existing taxi order data in the city. Then, the probability map of all geogrid crowd flow patterns could be constructed by combining nighttime remote sensing and land use data. At night, real-time monitoring is more difficult. When it is necessary to monitor or forecast the night crowd flows of a certain area, just input the spatial-temporal data into MG-STM as recent as possible, where we can then obtain the change of nighttime crowd flow patterns in the

concerned geogrid, and ensure the smooth implementation of policies such as closed-loop management in pandemic control and crime tracking.

## 6. Conclusions

Understanding and analyzing patterns of crowd flows is a crucial component of transportation planning and management. However, few studies have focused on the behaviors and regularity of nighttime crowd flows. Due to the suspension of public transportation at night, taxi orders are critical in capturing the features of nighttime crowd flows in a tourism city. This paper is the first to study patterns of night crowd flows in a tourism city, and combined with kinds of data resources. This paper uses Haikou which is a typical tourism city as an example and proposes the mixed geogrid spatio-temporal model (MG-STM) for effectively analyzing the patterns of nighttime crowd flows. This study draws the following conclusions:

1.  By incorporating land use data, the crowd flows obtained by taxi services are no longer limited to point-to-point analysis, which provides a baseline for future grid-based analysis in smart cities;
2.  Due to the high resolution of LJ1-01 in east Asia, the association with the LJ1-01 nighttime light image makes the update of night population density information more accurate. The traditional machine learning and deep learning methods are difficult to effectively extract spatio-temporal information because of multi-source and heterogeneous data. In this paper, the semi-supervised DCNMF and K-Means method is used to successfully complete the spatio-temporal dimension reduction and clustering. Then, the three types of data ((taxi orders, LJ1-01 nighttime light data and land use data)) are organically combined to explain the patterns of nighttime crowd flows from different angles.
3.  We also analyze the crowd flows' pattern changes in Haikou "before midnight" and "after midnight" and between downtown and citywide areas. It was found that the greater the lag in crowd flow in a certain area is (that is, the clustering of crowd flow change is "increase then decrease" or especially "decrease then increase"), the more prosperous and higher population density the area is, and the greater the land use patterns are closer to downtown residential and commercial areas. Thus, the prosperity of land use areas shows a high positive correlation with the lag of crowd flows. For every 5% increase in population density, the peak of crowd flows will be delayed by 30 min.

According to the patterns of nighttime crowd flows in urban land use areas, analyses of urban crowd flows and regional planning will become increasingly important for future city monitoring research and epidemic prevention. However, patterns of crowd flows based on urban land use types still remain insufficient. In future research, analysis will be conducted to detect the motivation of crowd journeys based on other crowd flow information, such as social media check-in data and mobile SIM card information. In addition, we will also carry out deeper research into more complex crowd flow situations, which could reveal the patterns of crowd flows in various land types during an entire day.

**Author Contributions:** Conceptualization, B.H. and W.Z.; methodology, B.H.; validation, W.Z.; formal analysis, B.H.; investigation, B.H.; writing—original draft preparation, B.H.; writing—review and editing, D.Z.; visualization, J.P.; supervision, C.C. All authors have read and agreed to the published version of the manuscript.

**Funding:** This research was funded by the National Key Research and Development Program of China (Grant No. 2021YFB3901300, 2018YFB0505304), National Precision Agriculture Application Project (Grant/Contract number: JZNYYY001), Beijing Municipal Science and Technology Project (Grant/Contract number: Z201100008020008).

**Institutional Review Board Statement:** Not applicable.

**Informed Consent Statement:** Not applicable.

**Data Availability Statement:** This study selects the taxi order data of Didi Company for Haikou from 1 May to 31 October 2018, data sources are non-public. The land use data of Haikou are obtained from http://www.dsac.cn/DataProduct/Detail/302406, and the resolution of these data is 100 m, we last accessed the link on 15 August 2020. In the research of nighttime light images, this study selects nighttime light data of LJ1-01 in Haikou city on the evenings of 4 September, 20 September 2018 (http://59.175.109.173:8888/index.html), we last accessed the link on 5 August 2020.

**Acknowledgments:** We gratefully thank the anonymous reviewers for their critical comments and constructive suggestions on the manuscript.

**Conflicts of Interest:** The authors declare no conflict of interest.

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
