# Peer review of "Patterns of Nighttime Crowd Flows in Tourism Cities Based on Taxi Data—Take Haikou Prefecture as an Example"

_remotesensing, doi:10.3390/rs14061413_

Round 1
Reviewer 1 Report
I do not agree with the following assumptions in the paper"
"The higher the nighttime light radiance level is, the greater the population density in the corresponding area"
We can speak about the artificial sources of light only in the context of the space and time level of human activity (the nighttime lights is only the proxy measure for the population density). I am not convinced that above thesis is true, especially in the context of the presented legend of map of land use which is categorically illogical (non-exclusive types): Residential areas (downtown, village i.e. built-up areas of urban and rural fabric) may include tourism resorts vs 'Tourism areas' (which also include natural areas as well hotels, POIs and Forests); additionally such generalization excludes transportation areas (e.g. airports, marine ports, highways) as well agricultural (farming). Moreover, 'Other' land use class my suggest unused land which is not true because may aggregate several other classes including inland water (which in turn may be included into 'Tourism' areas) as well sea beaches.
Assuming correctness of factorization method and classification of crowd flows (of taxi movement) using the proposed DCNMF method. There are no any population density data in paper compared to nighttime (artificial sources) light radiance and inferring about population density through percentage of categorized into three types night taxi flows is very limited and obvious.
Reviewer 2 Report
Review on the paper entitled “Patterns of nighttime crowd flows in tourism cities based on taxi data——take Haikou prefecture as an example”.
This paper investigates the patterns of crow flows in Haikou city based on nighttime light data and taxi order data provided by a ride-hailing company. The paper is very well-designed, -written, and -structured. The figures are of high quality. The topic is interesting and useful for cities that are popular tourist destinations therefore it may attract international attention.
Nevertheless, I have some remarks that the authors should address to improve the quality of the paper.
1) The authors focused on analysing the patterns of crowd flows in Haikou city whereas in the title of the paper they assert that Haikou prefecture was taken as an example. This issue is a bit complicated because Haikou is a prefecture-level city. The authors should clarify which administrative level was chosen for the analysis.
2) In line 72, the expression “tourist cities” can be read, and in other parts of the paper, the expression “tourism city” appears.
3) According to the authors, Haikou is a touristic destination. This assertion should be underpinned by tourism data (e.g., number of visitors). Furthermore, we know nothing about Haikou’s population and population density whereas the paper is somehow centred on these issues.
4) I’m a bit confused about the timeline of the research. In lines 128-130, on page 3, the authors write that “This study selects LJ1-01 nighttime light images of Haikou city on the evenings of September 4, September 20, and February 15…”. In the caption of Figure 2, it turns out that the nighttime light images come from 2018. Then in lines 153-154, on page 5, the following can be read: “Thus, this study selects the taxi order data of Didi Company for Haikou from May 1 to October 31…”. Do the taxi order data come from 2018 as well? Following this, from Table 1, we can conclude that the departure time data come from 2017. So, how does the timeline of the research work?
5) In lines 141-143, on page 4, the authors posit that “during the morning rush hour, the density of people traveling in the heatmap from 5 am to 9 am is significantly lower than normal crowd flow.” What is the “normal crowd flow” according to the authors? In addition, what does “traveling in the heatmap” mean?
6) The expression “MG-STM” first appears in line 298, on page 11, but the authors only reveal the meaning of the expression in line 445, on page 17. This should be corrected.
7) The authors write in lines 359-360 on page 13, that they ‘take the central urban area of Haikou as the research object, and collect corresponding data from the "China Haikou Yearbook" accordingly’. Firstly, what is the relationship between Haikou prefecture (prefecture-level city?), Haikou city and the central urban area of Haikou? Secondly, what is the date of the China Haikou Yearbook?
Reviewer 3 Report
The article is interesting and the problem is worth exploring, but it is recommended to complement the discussion and conclusion section with an indication of what is innovative in the presented solution in comparison to other such solutions (authors' contribution to the development of science). It would also be good to characterize the barriers that the authors faced in their experiments and to point out the advantages of the presented solution compared to other research in this area. In my opinion, the purpose of conducting the research is not entirely clear. From the research presented, we obtain information about the number of taxi passengers boarding and alighting in particular areas without knowing the motivation of their journey. In the discussion and conclusions, the authors should describe more precisely how to use the results of the research (e.g. how to use the results to monitor the flow of taxi passengers and pandemic mitigation). Information on travel flows between areas would be important.
Have the results been statistically verified?
In my opinion, it is reasonable to describe the relationship of the number of passengers getting in and out of taxis in an area with sociodemographic characteristics. Similarly, in the case of the dependence of the number of taxi passengers getting on and off taxis on the intensity of street lighting, although it is obvious that the most intensively lit areas are those with higher activity and the number of people moving (areas with the highest generation and absorption of trips and their surroundings, where the number of people travelling accumulates). It would be valuable to develop models describing these relationships to understand the results presented in the article.
Reviewer 4 Report
The authors of the submitted manuscript use a data dimensionality reduction (DCMNF) on taxi pick-up and drop-off data, gridded into 100 m2 blocks, followed by a k-means clustering algorithm to find patterns of nighttime public movement patterns in Haiku city, China. The authors use LJ1-01 nighttime lights data as a loose proxy of population density to further substantiate and interpret the observed commuting patterns against the population densities.
While the work involves an innovating data science-driven approach, there are several concerns that the authors need to address before this manuscript may be recommended for publication. Additionally, the authors should also address the concerns and comments of the other reviewers.
Please find the comments below:
- The LJ1-01 nighttime lights data has been used to proxy population density. But the coupling between population and light intensities is purely based on observation and not any quantitative analysis or validation. Several known issues with nighttime lights data could obscure the relationship between light intensity and population. E.g., commercial lighting results in high nighttime light intensity in business areas. But in reality, there may be very few people who may live at those places. The authors need to address these issues while using nighttime lights data. It would be better to include a fundamental quantitative analysis between population and nighttime lights values to translate the light data into a population layer before interpreting commuting patterns alongside population densities.
- Also, this paper's minimal use of remote sensing is based on observation rather than analysis. On line 382, the authors mention that they can use nighttime lights data to gather a population distribution within each grid. However, no quantitative analysis was done to create a relationship between nighttime light intensity and population (as mentioned in comment 1). Such application of nighttime lights data to infer population estimation is fairly common. Therefore, such limited use of remote sensing methods may also indicate that this paper may be well suited in a more GIS-focused journal. The authors are requested to address this issue.
- The authors explain the DCNMF and K-means methods in detail. But they do not do the same for overall MG-STM. The lack of a detailed description of the MG-STM method makes understanding the flow of paper difficult.
- Please mention the year of the datasets used in this study. Just the month and day is not sufficient. There may be a year-to-year difference among datasets that could have implications for interpretation.
- The authors assert that the taxi is an efficient way of capturing mobility patterns. However, taxi ridership may only capture a section of society that has the financial means to pay for taxis and exclude people who may want to use a bicycle or walk. Please address this potential issue that could bias your results.
- Figures are not clear, and the legends are hard to read. Figure 5 is missing a legend.
Round 2
Reviewer 1 Report
All my remarks have been taken into account in new version of paper.
Reviewer 3 Report
Thank you for improving the article according to my suggestions.
Reviewer 4 Report
Thank you for addressing the concerns and comments.